# Effect of Roller Levelling on Tensile Properties of Aluminum Sheets

**DOI:** 10.3390/ma16083001

**Published:** 2023-04-10

**Authors:** Dóra Harangozó, Imre Czinege

**Affiliations:** Department of Materials Science and Technology, Széchenyi University, 1 Egyetem tér, 9026 Győr, Hungary; harangozo.dora@ga.sze.hu

**Keywords:** roller levelling, mechanical model, linear hardening, change in flow stress

## Abstract

The straightening of sheets, bars and profiles plays an important role in many machining processes. The aim of sheet straightening in the rolling mill is to ensure that the deviation of sheets from flatness is within the tolerances specified in the standards or delivery conditions. There is a wide range of information available on the roller levelling process used to meet these quality requirements. However, little attention has been paid to the effects of levelling, namely the change in properties of the sheets before and after roller levelling. The aim of the present publication is to investigate how the levelling process affects tensile test results. The experiments have shown that levelling increases the yield strength of the sheet by 14–18%, while it decreases its elongation by 1–3% and hardening exponent by 15%. The mechanical model developed allows changes to be predicted, so that a plan can be made regarding roller levelling technology that has the least effect on the properties of the sheet while maintaining the desired dimensional accuracy.

## 1. Introduction

In roller levelling, a strip is forced through opposing rollers from a coil. The rollers bend the sheet in alternating directions in order to minimize the curvature. The dominant deformation process in sheet levelling is repeated bending, at the beginning of which elastic deformation occurs in the sheet and the stress associated with the elastic limit strain (*ε_e_*) of the outer fiber is just equal to the yield stress (*σ_y_*). As the sheet is moving forward, its radius of curvature decreases continuously, causing the strain of the outer fibers to increase and a plastic deformation to occur around them. When a pointed outer fiber of the sheet is bent on the opposite side of the symmetry plane of the two supporting rollers, the strain reaches its maximum (*ε_f_*). Upon leaving this position, when the tested point of the sheet arrives to the next roller, the direction of the stress and strain changes from tensile to compressive, subsequently passing the next rollers, the process is repeated in a cyclical manner [1]. This is illustrated in Figure 1a, which shows the arrangement of rollers and sheet. The graph indicates the strain of the lowest fiber of sheet. Figure 1b displays the stress and strain distribution in S2 and S3 sections; S2 is the first bending phase where the lower side of sheet is under tension and this will change to compression in the S3 section.

If the material exhibits isotropic hardening behavior, the tensile and compressive yield strengths are the same, but if the material model is characterized by kinematic hardening, the compressive yield strength is smaller than the tensile yield strength due to the Bauschinger effect.

The roller levelling process has been analyzed in several publications. In the earliest research, mechanical models were established; some good examples can be found in [1,2,3,4,5,6,7,8,9]. Ref. [1] is a milestone in the development of roller levelling models. It gives a general introduction to the process and provides an analytical solution for calculating forces, moments, residual stresses and flatness. Ref. [2] applied the residual curvature model to analyze the effect of thickness, incoming flatness and yield strength on residual stresses and curvature at long bar straightening. The authors of [3] performed a uniaxial tension–compression bending test to characterize material properties under cyclic loading and used the experimental results to model the roller straightening process. The novelty of [4] was the combination of analytical and FEM model to describe the process parameters. Refs. [5,6,7] considered the initial sheet curvature and applied an iterative method for calculating the effective contact points between rolls and sheet. Refs. [8,9] provided a solution for calculating residual stresses after levelling. The authors of publication [9] used classical bending theory of bars for analysis with reasonable accuracy. The aim of these analyses is to optimize the process parameters. As a result, the authors determined the bending moment and forces acting on the rollers, while the curvature of the outgoing sheet and the residual stresses were evaluated as quality parameters. Finite element calculations are increasingly used to analyze the entire levelling process; several examples can be found in the literature [10,11,12,13,14,15]. Based on model calculations, it is possible to control the straightening device by modifying the force on the rollers, thus minimizing the curvature and residual stresses [16].

Of particular interest in roller levelling research is the publication [17], which attempts to predict the curvature of the straightened sheet based on input parameters using artificial intelligence tools, namely artificial neural network (ANN)-based modelling. This method is known in materials science; for example, the publication [18] gives an extensive overview of the results. From the publication [17], it can be concluded that a neural network derived from a training set of 600 measured values gives much better results for the output parameters of the validation set than any other mathematical method. However, to apply this approach, a significant database size is required. In another paper, the authors of the aforementioned publication used a multivariate linear regression model to characterize the process of roller levelling [19], which also produced favorable results.

Two basic modelling techniques have been developed for roller levelling. The model based on the classical bending theory of bars has symmetrical arrangement, with the sheet laying on the top of two neighboring bottom rollers and the upper roller transfers the load in the direction of the vertical symmetry line. In this case, the relationships developed for bending can be applied without modification to determine the stresses and deflections, as shown, for example, in [8]. The other model takes into account that the sheet is initially curved when it is fed to the levelling rollers and then becomes curved due to further straightening, so in reality the support points will not be exactly at the top or bottom of the rollers. This changes the position and direction of the acting forces, and therefore a complicated iteration procedure should be used to determine the final position of the contact points [5,8]. Although calculations show that the position of the contact points does not change significantly compared to the symmetric arrangement, they deviate by around 3° from the vertical axis on the first rollers and on subsequent rollers they fluctuate by around 1–2° [6]. Furthermore, it can be seen from the publication [11] that the force values between two adjacent rollers differ by only a few percent, so the simplified geometric model can be applied to certain tasks, keeping the symmetric geometric arrangement and assuming that the moments M_i−1_, M_i_ and M_i+1_ on three adjacent rollers are the same.

Besides the optimization of the process parameters, relatively few publications have addressed the change in tensile properties of the roller levelled sheet. However, such a change can certainly happen, since inhomogeneous plastic deformation occurs during roller levelling and residual stresses in the sheet are also generated. In [20], the authors investigated how the plastic zone affects the springback experienced after V-bending with a tool angle of 90°. They found that varying the ratio of the plastic zone between 60 and 85% changes the bending angle by approximately 3% and increases the bent radius of the sheet by ~30% after springback. The publication [21] deals with the variation of hardness distribution of 4 mm thick HSLA steel, measurements taken show that the hardness of a roller levelled sheet changes little compared to the one that is not levelled.

Based on the literature review, it can be concluded that research on roller levelling has shown significant achievements; the mechanical and finite element models fit well to the experimental results. However, there are very few publications on the effect of levelling on the mechanical properties of the sheet, despite the demand from application aspects of products. It is known from workshop practice that roller levelling, for example, leads to a reduction in tensile elongation, which in some cases risks the acceptability of the product. Therefore, the main objective of this publication is to analyze the relationship between roller levelling parameters and the mechanical properties determined by tensile testing.

## 2. Materials and Methods

### 2.1. Materials

In connection with the aluminum rolling mill, several experiments have been carried out, the aim of which was to study the effect of roller levelling on the properties of products. In order to detect changes, samples of some products were taken both before and after roller levelling. In the following charts and tables, the measured values before roller levelling will be labelled with WL (Without Levelling), while the Roller Levelled samples will be labelled with RL. The sheet grades tested belong to the AlMg3 or AlMg4.5 aluminum alloy family and their basic chemical composition is given in Table 1.

Four process variants were produced from a single batch of AlMg3 alloy, combining cold forming and annealing in the final processing. The sequence of cold rolling (CR), annealing (A) and stress releasing (SR) operations is visualized in the next scheme (Table 2). Annealing was carried out at 320 °C for 2 h, the amount of deformation during cold rolling is indicated in the scheme. Stress release is different from annealing as it means a heat treatment at lower temperature (270 °C), causing strong decrease in residual stresses and dislocation density. Without stress releasing or annealing, the cold rolled structure has lower formability. AlMg4.5 sheet was produced in only one process variant.

For the tests, three specimens were produced from each batch of rolling direction (RD), diagonal direction (DD) and transverse direction (TD). The tests were performed on an Instron 5582 tensile test machine equipped with a video extensometer for measuring longitudinal and transversal strain.

### 2.2. Mechanical Model

The roller levelling equipment with the manufacturer providing information for this research consists of 17 rollers, as shown in Figure 2a. The size of the rollers and their distance from each other is shown in the figure. The mechanical model developed is based on the simplified geometric arrangement briefly described above. As mentioned in the introduction, the symmetrical arrangement, the equality of bending moments and supporting forces do not cause significant deviations from the real conditions. Furthermore, the main objective of this analysis is to establish relationships between the roller levelling parameters and sheet properties; furthermore, the simplified model is more favorable for this purpose.

When building the model, it is sufficient to consider the layout shown in Figure 2b for symmetry reasons. The figure is not scaled for better clarity, as the diameter of the rollers is much larger in reality while maximum deflection *s* is smaller. The center of the *x-w* coordinate system is in the vertical symmetry plane of the left-hand roller, half a sheet thickness from the roller surface, so the radius between the sheet center line and the roller center is *R_k_* = *R_r_* + *t*/2. Since the distance between the bottom two rollers is *P*, the horizontal distance between the upper/lower rollers in the figure is *P/*2. The inflection point of the curved sheet is at a distance *x* = *P/*4, which indicates the *M* = 0 location of bending moment. The variation of bending moment along *x* is shown in the figure and the supporting forces on the rollers are also marked symbolically. It should be noted that the model presented describes the deformation of the sheet using equations known from the mechanics of beams. In roller levelling literature, intermesh is often used as a characteristic geometrical parameter; it is the vertical distance between the upper point of the lower roller and the lowest point of the upper roller. Assuming that this quantity is positive if the bottom point of the upper roller is above the plane connecting the top of the lower rollers, there is a relationship between maximum deflection *s*, sheet thickness *t* and intermesh *i*, which is *s* = *t − i*.

From the mechanics of bended beams, it is known that the curvature can be calculated by *κ* = *M/IE*, where *M* is the bending moment, *I* is the moment of inertia and *E* is the elastic modulus. The curvature of a line can be written as a function of the first and second derivatives, as shown in Equation (1). Since the deflection of the curved sheet is small relative to the length of the beam, the square of the first derivative in the denominator can be taken to be approximately 0, so that the curvature will be equal to the second derivative.


(1)
$$\kappa =\frac{w^{\prime \prime }}{{\left(1+w^{\prime 2}\right)}^{3/2}}\sim w\prime \prime$$


Substituting this into the relation for the curvature, the equation for the second derivative of the center line of the curved sheet can be described by the expression *IEw*″*(x)* = *M*. Integrating this with respect to *x* gives the first derivative *w*′*(x)*, which is the slope of the centerline, and further integration gives the function *w(x)* for the shape of the sheet centerline. The model shown in Figure 1b must be decomposed into two parts, the section from *x* = 0 to *P/4* and the section from *P/*4 to *P/*2. The equations for each part are shown in the following denoted by Equations (2)–(4) and Equations (5)–(7).

For *x* = 0…*P*/4
(2)$$IEw^{''}=M=\frac{F}{2}\left(\frac{P}{4}-x\right),$$
(3)$$IEw^{\prime }=\frac{F}{2}\left(\frac{P}{4}x-\frac{x^{2}}{2}\right)+C_{1},$$
(4)$$IEw=\frac{F}{2}\left(\frac{P}{8}x^{2}-\frac{x^{3}}{6}\right)+C_{1}x+C_{2},$$
boundary conditions at *x* = 0:

*w*′(0) = 0; from this *C*_1_ = 0

*w*(0) = 0; from this *C*_2_ = 0

For *x* = *P*/4…*P*/2; be *x** = *x* − *P*/4
(5)$$IEw^{\prime \prime }=M=\frac{F}{2}x^{*},$$
(6)$$IEw^{\prime }=\frac{F}{2}\left(\frac{x^{*2}}{2}\right)+C_{1},$$
(7)$$IEw=\frac{F}{2}\left(\frac{x^{*3}}{6}\right)+C_{1}x+C_{2},$$
boundary conditions at *x** = 0:

*w*’(*x** = *P*/4) = 0; from this *C*_1_ = −*FP*^2^/64

*w*(*x** = 0) = *s*/2; from this *C*_2_ = *IEs*/2

By substituting the *C*_1_ and *C*_2_ constants, *w, w*′ and *w*″ can be calculated. However, for roller levelling, the most important thing to know is the shape of sheet as a function of maximum deflection *s*, using the relationship between curvature and radius of curvature *(κ =* 1*/R_k_)* and the maximum deflection of supported beams *(FP/8IE)*.


(8)
$$\frac{1}{R_{k}}=\frac{M}{IE}=\frac{FP}{8IE},$$



(9)
$$\frac{s}{2}=\frac{F{\left(P/4\right)}^{3}}{48IE},$$


From Equations (8) and (9), the *F/IE* quotient can be expressed to obtain a relationship which is independent of sheet material, size and force:


(10)
$$\frac{F}{IE}=\frac{8}{R_{k}P}=\frac{192s}{P^{3}},$$


Calculating boundary conditions using Equations (8)–(10) and substituting back into Equations (2)–(7) gives Equations (11)–(14) for the deflection of the center line of the sheet and the curvature as a function of *x*, *s* and *P*. From these equations, the strain *ε_f_* in the outer fiber according to Equation (15) and the radius of minimum curvature *R_kmin_* are obtained, as shown in Equation (16). Using Equations (15) and (16), the elongation of the outer fiber as a function of sheet thickness *t* and maximum deflection *s* can be written, as shown in Equation (17). Knowing these, the plastic zone ratio *η* can also be expressed in terms of the geometry of the roller leveler equipment, as shown in Equation (18). In this equation, the elastic strain limit is *ε_e_ = σ_y_/E*, where *σ_y_* is the yield strength and *E* is the elastic modulus. The basic equations of levelled sheet are the following:

For *x* = 0…*P*/4


(11)
$$w=-\frac{192s}{P^{3}}\left(\frac{Px^{2}}{16}-\frac{x^{3}}{12}\right),$$



(12)
$$w^{\prime \prime }=\frac{1}{R_{k}}=\frac{192s}{P^{3}}\left(\frac{P}{8}-\frac{x}{2}\right),$$


For *x* = *P*/4…*P*/2


(13)
$$w=\frac{192s}{P^{3}}\left(\frac{{\left(x-P/4\right)}^{3}}{12}-\frac{P^{2}\left(x-P/4\right)}{64}\right)-\frac{s}{2},$$



(14)
$$w^{\prime \prime }=\frac{1}{R_{k}}=\frac{96s}{P^{3}}\left(x-\frac{P}{4}\right),$$



(15)
$$\mathrm{strain}\;\mathrm{of}\;\mathrm{outer}\;\mathrm{fibre}:\;\varepsilon_{f}=\frac{t}{2R_{k}},$$



(16)
$$\mathrm{minimum}\;\mathrm{curvature}\;\mathrm{radius}:\;R_{kmin}=\frac{P^{2}}{24s},$$



(17)
$$\mathrm{substituting}\;R_{\mathrm{kmin}}\;\mathrm{to}\;(15):\;\varepsilon_{f}=\frac{12st}{P^{2}},$$



(18)
$$\mathrm{ratio}\;\mathrm{of}\;\mathrm{plastic}\;\mathrm{zone}:\;\eta =1-\frac{\varepsilon_{e}}{\varepsilon_{f}}=1-\frac{P^{2}}{12st}\varepsilon_{e},$$


The resulting equations can be used to plot the quantities calculated from the geometric characteristics of the roller levelling equipment shown in Figure 2a, which are illustrated in Figure 3 (assumed sheet thickness *t* = 2.5 mm). Figure 3a–c show the variation of the deflection *w* (Figure 3a), the strain of the outer fiber *ε_f_* along the length (Figure 3b) and the radius of curvature *R_k_* (Figure 3c) as a function of the coordinate *x* in the direction that the sheet travels. The latter quantity is infinite at *x* = *P/*4, and therefore cannot be plotted continuously. Figure 3d shows the plastic zone ratio *η* calculated from relation (18), from which it can be seen that the ratios around 0.95, typical in roller levelling practice, are obtained at 3.5 mm maximum deflection for these geometries.

The reality of the developed equations is confirmed by the fact that the levelling parameters of a 4 mm thick sheet with a yield strength of 500 MPa on a 14-roller straightening line calculated using the initial data given in [5] are in practically full agreement with the results of the model calculation. A similar agreement was shown by comparing the data obtained using the finite element technique in [11] with the model calculations.

### 2.3. Material Model

Using Equations (11)–(18) and an appropriate material model, the yield strength increment due to roller levelling can be predicted. For this purpose, the stress–strain curve of the sheet without levelling must be known, since the material model of the initial sheet must be calculated from it. The first analysis of suitable material models can be found in [1], where two hardening models, linear and exponential, were compared. The authors of this study verified that both are suitable for calculations. The linear elastic-plastic model was used in the publications [5,11,13], while the authors of [3,15] preferred the exponential model. Considering these results, several models were analyzed in our preliminary studies (linear, power and exponential hardening), but since the maximum plastic strain of the sheet during roller levelling is about 4%, the linear elastic-linear strain hardening model proved to be the most suitable.

The elastic section is described by the equation *σ_e_* = *Eε_e_*, and the equation for the linear hardening section is given by *σ* = *σ*_0_
*+ Dε*, where *σ_e_* is the elastic stress, *E* is the elastic modulus, *ε_e_* is the elastic strain, *σ_0_* is the intercept and *D* is the slope of line (Figure 4a). The numerical values in Figure 4a refer to a test presented later. Using this material model, it is possible to construct the stress distribution in the elastic and plastic zones for a specific roller levelling process as shown in Figure 4b as a function of actual sheet thickness *t_y_*, where 0 ≤ *t_y_* ≤ *t/*2 = 1.25 mm. For symmetry reasons, only the stresses of one side of the sheet are shown here; this is due to the assumption of the isotropic hardening model.

The yield stress in a sheet without levelling is equal to *σ_y_* over the whole cross-section, and the stress distribution after roller levelling starts with yield stress and keeps this value up to the elastic limit, then from this point it increases linearly. Obviously, the measured yield stress during the tensile test of the levelled sheet is modelled by the average stress, so the ratio of the average stress to the initial yield stress characterizes the calculated yield stress increment. The equations describing the elastic and linear hardening section define two lines; their intersection gives the elastic yield strength in the form *ε_e_* = *σ*_0_*/(E* − *D)*. Knowing the surface strain *ε_f_* from Equation (15), the elastic strain and the material model parameters, the average stress on the cross-section can be expressed by the following equation:


(19)
$$\bar{\sigma }=\sigma_{e}+\eta \left(D\frac{\varepsilon_{f}+\varepsilon_{e}}{2}+\sigma_{0}-\sigma_{e}\right),$$


The quantities in Equation (19) can be expressed as a function of maximum deflection *s* and sheet thickness *t* to represent the yield strength increment due to roller levelling.

This is illustrated in Figure 5, which shows that the average stress developed in this model is a linearly increasing function of maximum deflection, so that the greater the deformation caused by levelling, the more intense the increase in yield strength. The figure also shows the evolution of the maximum stress, which confirms that there is significant hardening on the surface of the sheet.

## 3. Results

### Tensile Test Results

The engineering stress–strain curves of some specimens recorded during the tensile tests are shown in Figure 6, while Table 3 shows the average values and the ratio of the tensile test characteristics of all the samples roller levelled and without levelling, indicating the extent of increase or decrease caused by levelling.

Important conclusions can be drawn from the presented stress–strain curves on a visual basis (Figure 6). It was verified by model calculations that roller levelling causes a plastic deformation of 1–4% on the surface of the sheet, which is equivalent to the ratio of plastic zone η covering 70–95% of the total width of the sheet. The strain hardening caused by plastic deformation results in an increase in yield strength of levelled samples compared to samples without levelling. However, the tensile strength does not change significantly because the deformation is very small. It is also noticeable that the elongations at the break of the levelled specimens are in all cases smaller than the reference values without levelling.

## 4. Discussion

### 4.1. Analysis of Tensile Test Results

The findings derived from engineering stress–strain curves (Figure 5) have been confirmed by a detailed study of the measured values displayed in Table 3. From the comparison of roller levelled and not levelled values (RL/WL), it can be seen that among the strength properties, the yield strength *σ_y_* increased on average by 14% for AlMg3 and 18% for AlMg4.5, while the ultimate tensile strength *σ_UTS_* changed by only approximately 1%. The numerical values show that the increase in yield strength mainly affects the hardening exponent *n*, it decreases by approximately 15% for both sheets. This can be explained by the fact, that the roller levelled specimens have higher yield strength/tensile strength ratio and therefore lower hardening exponent. The uniform elongation *ε_u_* and tensile strain at break *ε_TS_* indicates a 3–4% decrease after roller levelling. The average normal anisotropy *r* shows no change in the average, but the scatter of measured values is significant, whereas the planar anisotropy Δ*r* increases with levelling. These measurement results are illustrated in the graphs in Figure 7, where Figure 7a shows the yield strength comparison for specimens in the rolling direction and Figure 7b–d compare the corresponding quantities for roller levelling and without levelling sheets. The points above the *x* = *y* line indicate that the illustrated property is higher for the levelled sheet (e.g., yield strength *σ_y_* in Figure 7b, and the points below the line indicate a decrease due to levelling (e.g., *n* in Figure 7d).

Particular interest should be given to the relative change of tensile test parameters related to properties without levelling, as these also highlight the specific effects of roller levelling. This is shown in the graphs in Figure 8, based on data from experimental samples of the AlMg3 sheet.

Figure 8a–c show the ratio of yield strength, elongation and hardening exponent for levelled and not levelled specimens as a function of the without levelling property. As demonstrated in Figure 8a, a larger initial yield strength is associated with a smaller increment, so that the hardened samples will continue to harden less than the annealed versions. The change is inversely true for uniform and tensile elongation, where the greater the elongation of the not levelled (WL) specimen, the more pronounced the decrease in elongation. The same is true for the hardening exponent. Figure 8d highlights only the yield strength and hardening exponent ratios of the four AlMg3 samples for values measured in the rolling direction. The bar chart (Figure 8d) clearly demonstrates that for hardened and annealed samples, the smaller yield strength versions (AlMg3/2 and AlMg3/4) show a larger change compared to higher strength (AlMg3/1 and AlMg3/3) sheets. This phenomenon can be explained by the non-linear nature of strain hardening. Roller levelling causes plastic deformation which increases the yield strength and reduces elongation. When this effect is applied to an annealed sheet with a low yield strength, the increase in yield strength is greater than when the same effect is applied to a strain hardened sheet due to the greater slope of the initial part of the hardening curve. Conversely, an annealed sheet with greater uniform and tensile elongation will lose a greater proportion of its ductility after roller levelling than a previously strain hardened sheet.

### 4.2. Comparison of the New Mechanical Model with Experimental Results

Selecting a sample from the tensile test results, the parameters *σ_0_* and *D* of the linear hardening material models can be determined, as well as the additional quantities needed to calculate the average stress using Equation (19). According to the information received from the manufacturer, the maximum deflection during roller levelling for the tested sheets was *s* = 2.16 mm. By substituting this into Equations (15)–(19), the average stress can be calculated, and the error can be determined by comparing it with the measured yield strength of the roller levelled specimen.

The numerical results are given in Table 4 and the comparison of the measured and calculated values is given in Figure 9. The measured and calculated yield stresses have a small scatter compared to the *x* = *y* straight line in red; the magnitude of the errors is given in the table.

The calculated results based on the presented model corroborate relatively well with the measured yield strengths and with an average absolute error of 3.42%, so the increase in σ_y_ can be calculated using the tensile test results before levelling from the planned process parameters. At the same time, the changes in tensile properties caused by levelling can be predicted. From Figure 4, it can also be concluded that levelling parameters with as small a deflection as possible should be chosen, as this will reduce the inhomogeneous deformation and increase yield strength.

## 5. Conclusions

The changes in the properties of the roller levelled and not levelled specimens determined by tensile tests are summarized below:The tested roller levelling process caused an average increase in yield strength of 14% for AlMg3 and 18% for AlMg4.5, while tensile strength changed by less than 1%.The theoretical model developed on the basis of levelling parameters and initial sheet properties correctly describes the increase in yield strength due to roller levelling calculated from the inhomogeneous deformation of the sheet, and this was clearly confirmed by comparison with experimental results.The value of the hardening exponent decreased by approximately 15% for both sheets, while the average normal anisotropy did not change.With regard to the change in the property ratios of levelled and not levelled samples, it was found that a higher yield strength is associated with a lower increase ratio, i.e., the samples hardened during production will suffer less increase in yield strength than the annealed versions.This phenomenon is inversely related to the uniform elongation, the elongation at break and the hardening exponent. The greater the elongation and hardening exponent of the not levelled sample, the higher their relative decrease.

## Figures and Tables

**Figure 1 materials-16-03001-f001:**
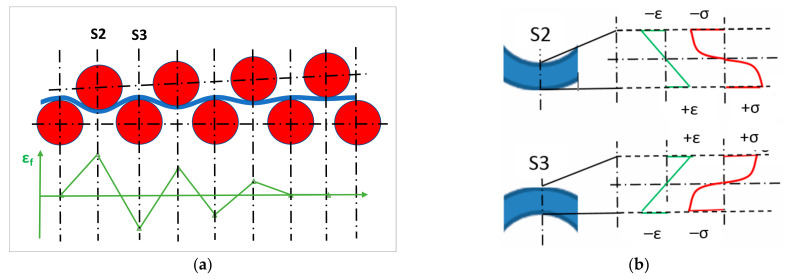
Illustration of roller levelling process; (**a**) simplified arrangement of rollers and sheet; (**b**) stress and strain distribution of two characteristic sections.

**Figure 2 materials-16-03001-f002:**
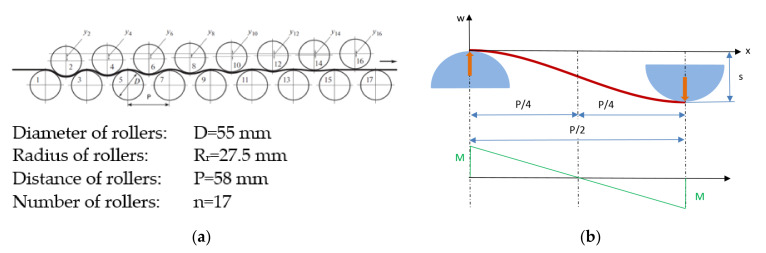
Roller levelling equipment and model parameters; (**a**) roller levelling line; (**b**) dimensions, forces and moment.

**Figure 3 materials-16-03001-f003:**
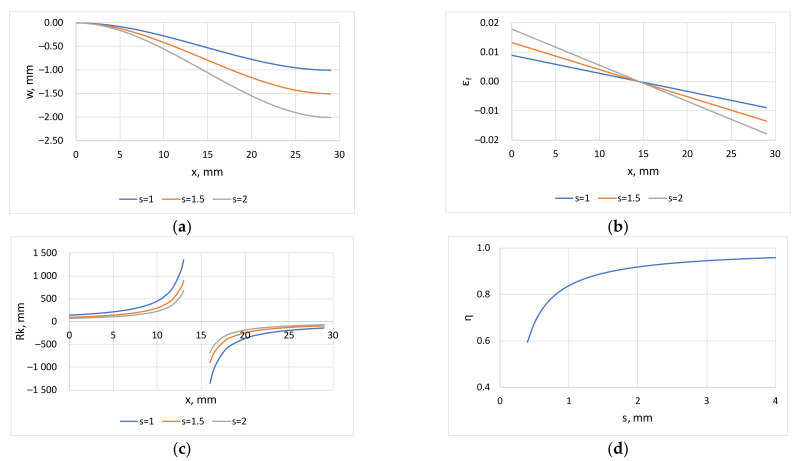
Characteristic parameters of roller levelling; (**a**) length deflection (Equations (11) and (13)); (**b**) strain of outer fiber (Equation (15)); (**c**) radius of curvature (Equation (16)); (**d**) ratio of plastic zone (Equation (18)).

**Figure 4 materials-16-03001-f004:**
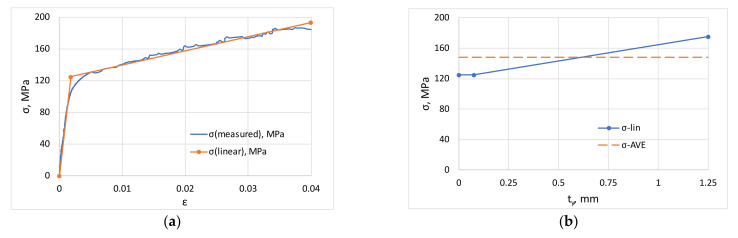
Material model and stress distribution (**a**) linear elastic-linear hardening model; (**b**) stress distribution along thickness.

**Figure 5 materials-16-03001-f005:**
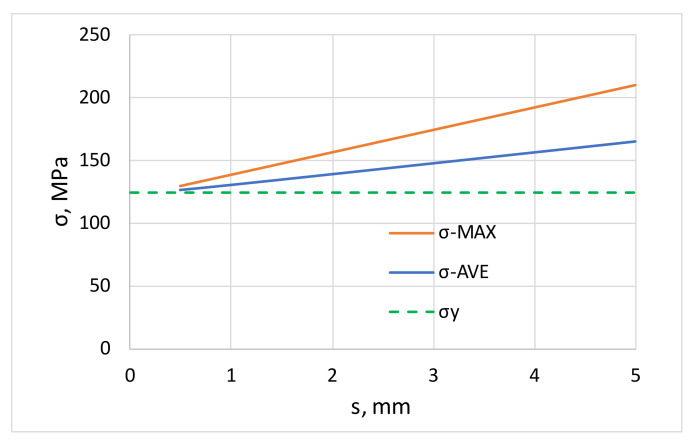
Increase in yield stress and maximum stress due to roller levelling as function of deflection.

**Figure 6 materials-16-03001-f006:**
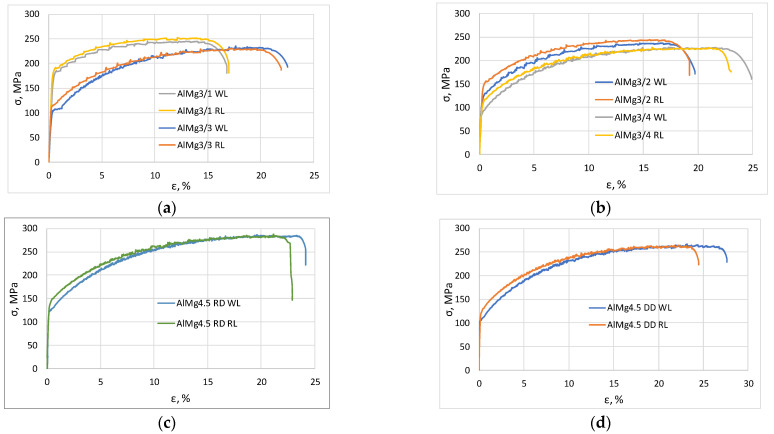
Typical tensile test results; (**a**) AlMg3/1 and AlMg3/3; (**b**) AlMg3/2 and AlMg3/4; (**c**) AlMg4.5 Rolling direction; (**d**) AlMg4.5 Diagonal direction.

**Figure 7 materials-16-03001-f007:**
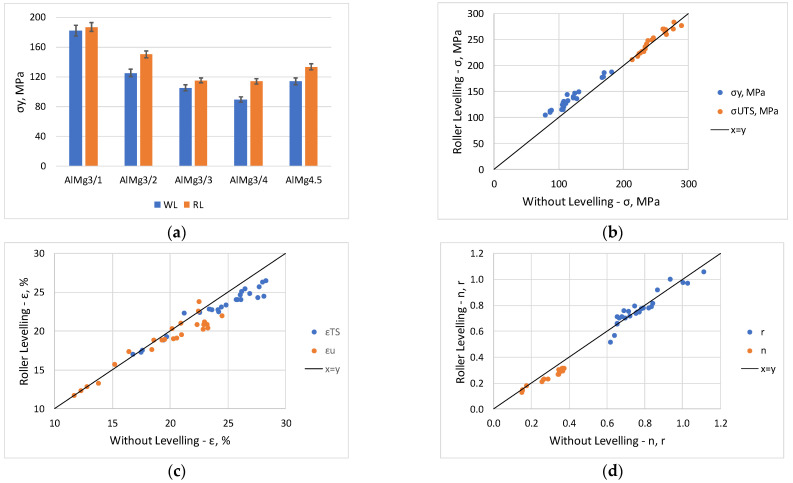
Comparison of tensile test results (**a**) yield strength comparison of AlMg3 and AlMg4.5; (**b**–**d**) further comparison of tensile test parameters.

**Figure 8 materials-16-03001-f008:**
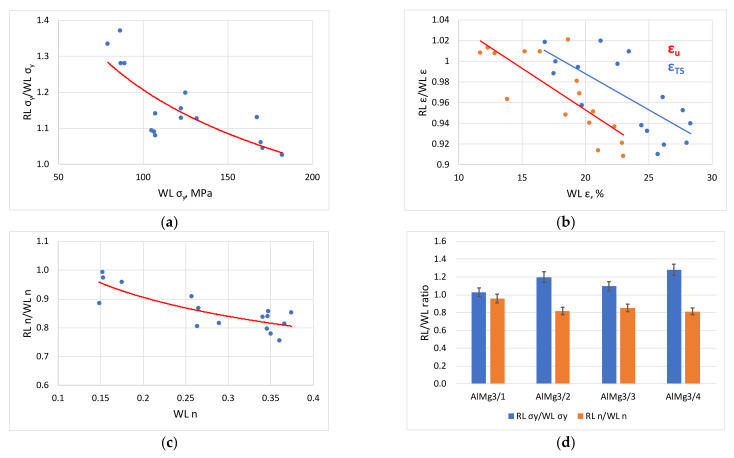
Relative change in tensile test parameters (**a**–**c**) the ratio of yield strength, elongation and hardening exponent for levelled and not levelled specimens; (**d**) yield strength and hardening exponent ratios for AlMg3.

**Figure 9 materials-16-03001-f009:**
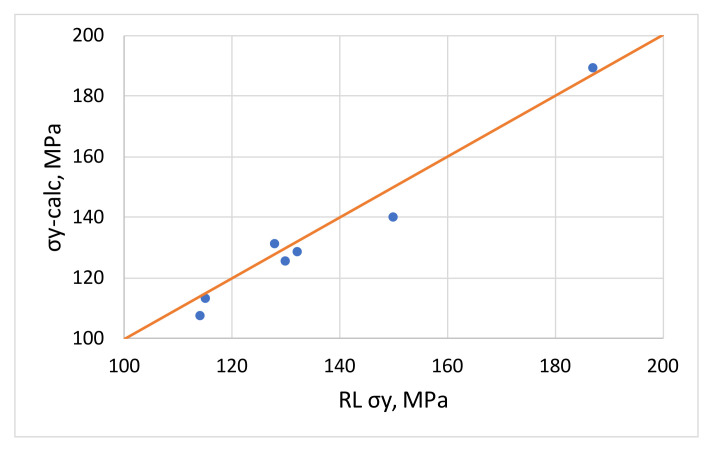
Comparison of calculated and measured yield stress values.

**Table 1 materials-16-03001-t001:** Main alloying elements of tested sheets.

Alloys	Mg (wt.%)	Fe (wt.%)	Si (wt.%)	Mn (wt.%)
AlMg3	3.30	0.11	0.05	0.33
AlMg4.5	4.69	0.11	0.09	0.25

**Table 2 materials-16-03001-t002:** The sequence of cold rolling, annealing and stress releasing operations.

Cold rolling		CR 15%	A 320 °C/2 h	CR 5%		AlMg3/1
		CR 15%	A 320 °C/2 h	CR 5%	SR 270 °C/2 h	AlMg3/2
		CR 20%	A 320 °C/2 h			AlMg3/3
	A 320 °C/2 h	CR 20%	A 320 °C/2 h			AlMg3/4

**Table 3 materials-16-03001-t003:** Results of tensile tests—average values of all specimens.

AlMg3	σ_y_, MPa	σ_UTS_, MPa	ε_u_, %	ε_TS_, %	n	r
AVE (WL)	122.26	231.94	18.00	23.09	0.28	0.75
AVE (RL)	139.34	234.52	17.43	22.30	0.24	0.75
RL/WL	1.14	1.01	0.97	0.96	0.85	0.99
**AlMg4.5**						
AVE (WL)	111.67	269.67	22.52	25.88	0.360	0.818
AVE (RL)	132.11	271.33	21.44	24.06	0.309	0.831
RL/WL	1.18	1.01	0.95	0.93	0.86	1.02

**Table 4 materials-16-03001-t004:** Comparison of measured and calculated yield strength values.

S = 2.16 mm	RL σ_y_	σ_y_-Calc	Error %
AlMg3/1 RD	187	189.24	1.20
AlMg3/2 RD	150	140.37	−6.42
AlMg3/3 RD	115	112.46	−2.21
AlMg3/4 RD	114	107.60	−5.61
AlMg4.5-RD	128	131.41	2.66
AlMg4.5-DD	132	128.58	−2.59
AlMg4.5-TD	130	125.74	−3.28

## Data Availability

Data is unavailable due to privacy.

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
