# Peer review of "Effect of Roller Levelling on Tensile Properties of Aluminum Sheets"

_materials, 2023, doi:10.3390/ma16083001_

Round 1

Reviewer 1 Report

Dear editors:

The paper has been assessed. I have raised some points which require clarification and amendments are required. I am enclosing these comments below.

In this paper, in order to investigate the change in properties of the sheets before and after roller levelling,the authors develop the model and gain the mechanical properties of materials. They found that the tyield strength of 14% for AlMg3 and 18% for AlMg4.5, tensile strength only changed by less than 1%, and the value of the hardening exponent decreased by approximately 15% for both sheets.

This paper is suitable for publication if the following problems are properly handled:

 1. The authors should give the result in abstract.

2. The error bar should be added in Figures ,such as Figure6(a).

3. It seems good to merge the two parts of Results and Discussion.

4. The authors should explain the difference between the no subsequent heat treatment and stress release heating.

5. Why the authors only gave one process variant AlMg4.5 sheet? 

Reviewer 2 Report

The article is devoted to the study of straightening sheet metal from aluminum alloys. The article is written competently, has scientific novelty and practical significance. I have a few comments that I think will be useful for improving the quality of the paper and publishing it in a scientific journal.

1. In the Introduction, when analyzing studies performed earlier on flattening of rolled products, it is indicated "The roller leveling process has been analyzed in several publications. In the earliest research, mechanical models were established, some good examples can be found in references [1] and [2 -10]." I recommend giving more information on each article. What exactly was studied in each work?

2. In 2.1 Materials, a description of the modes of processing of the studied samples is given. It is difficult to perceive this information from the text when reading. I recommend giving a graphical description of the modes of obtaining the samples under study. In the form of graphs and drawings, this information will be perceived by the reader faster.

3. I recommend the authors to add another figure explaining the distribution of elastic and plastic deformations, compressive and tensile stresses across the thickness of the sheet with alternating deformation depending on the bending diameter.

4. The graphs in Figure 2, in my opinion, are incomprehensible to the reader without drawings explaining what happens to deformations and stresses across the thickness of the sheet when the radius of curvature of the rolled product and the number of rollers change.

Reviewer 3 Report

1- The first paragraph of introduction does not have references.

2- It is recommended to clarify the first paragraph of introduction by using a schematic figure.

3- The nobility of the work is presented at the end of introduction.

4- Use the symbol for each type of experimental condition which identifies the condition of rolling, its strain, and kind of annealing.

5- Considering "Several models have been analyzed in preliminary studies, but since the maximum plastic strain of the sheet during roller levelling is around 4%, the linear elastic-linear strain hardening model has proved to be the most suitable.“,  please mention some of these models in the literature review.

6- Considering “The bar chart (Figure 7(d)) clearly demonstrates that for 293 hardened and annealed samples, the smaller yield strength versions (AlMg3/2 and 294 AlMg3/4) show a larger change compared to higher strength (AlMg3/1 and AlMg3/3) 295 sheets.”, please explain the reasons of these changes.

Round 2

Reviewer 2 Report

The authors have corrected the article taking into account my comments. I recommend the article for publication in this version.

Reviewer 3 Report

Dear authors,

The revised version can be accepted for publication in the journal.